# Common Secondary and Tertiary Structural Features of Aptamer–Ligand Interaction Shared by RNA Aptamers with Different Primary Sequences

**DOI:** 10.3390/molecules24244535

**Published:** 2019-12-11

**Authors:** Muslum Ilgu, Shuting Yan, Ryan M. Khounlo, Monica H. Lamm, Marit Nilsen-Hamilton

**Affiliations:** 1Roy J Carver Department of Biochemistry, Biophysics and Molecular Biology, Ames, IA 50011, USA; rkhounlo@iastate.edu (R.M.K.); marit@iastate.edu (M.N.-H.); 2Aptalogic Inc., Ames, IA 50014, USA; 3Department of Biological Sciences, Middle East Technical University, Ankara 06800, Turkey; 4Department of Chemical and Biological Engineering, Iowa State University, Ames, IA 50011, USA; syan@iastate.edu (S.Y.); mhlamm@iastate.edu (M.H.L.)

**Keywords:** neomycin-B RNA aptamer, aminoglycoside, 2-aminopurine (2AP), molecular dynamics, isothermal titration calorimetry

## Abstract

Aptamer selection can yield many oligonucleotides with different sequences and affinities for the target molecule. Here, we have combined computational and experimental approaches to understand if aptamers with different sequences but the same molecular target share structural and dynamical features. NEO1A, with a known NMR-solved structure, displays a flexible loop that interacts differently with individual aminoglycosides, its ligand affinities and specificities are responsive to ionic strength, and it possesses an adenosine in the loop that is critical for high-affinity ligand binding. NEO2A was obtained from the same selection and, although they are only 43% identical in overall sequence, NEO1A and NEO2A share similar loop sequences. Experimental analysis by 1D NMR and 2-aminopurine reporters combined with molecular dynamics modeling revealed similar structural and dynamical characteristics in both aptamers. These results are consistent with the hypothesis that the target ligand drives aptamer structure and also selects relevant dynamical characteristics for high-affinity aptamer-ligand interaction. Furthermore, they suggest that it might be possible to “migrate” structural and dynamical features between aptamer group members with different primary sequences but with the same target ligand.

## 1. Introduction

Aptamers are valuable as in vivo synthetic riboswitches and as in vitro sensors. They are useful subjects for studies of small molecule interactions with RNAs, especially when physical structural information (such as NMR or X-ray crystallographic data) is available. However, there are few solved structures of aptamer–small ligand complexes and even fewer of unoccupied aptamers. Knowledge of the dynamical aspects of aptamer–small molecule ligand interactions is also largely absent. To fill these gaps, computational renderings of the dynamics of molecular interactions are being increasingly adopted as a means of expanding our understanding of RNA-small molecule functional complexity and dynamics [1,2,3,4].

Aptamers are particularly challenging for understanding structure because they are selected in such variety. A single SELEX (Systematic Evolution of Ligands by EXponential Enrichment) experiment against a target molecule can yield many aptamers, each with a different primary sequence. Although one or two examples might be fully analyzed biophysically, structures for most aptamers are not likely to become known with atomic resolution. The aptamer examples chosen for biophysical analysis may not be the best choices for application as a riboswitch or a sensor. Yet, understanding the relevant structural features and dynamics by which an aptamer interacts with its ligand can provide invaluable guidance in developing aptamers as sensors for diagnostics. To address this problem, and based on our previous studies, we posed the hypothesis that the ligand plays a significant role in driving both aptamer structure and dynamics. If this is true then aptamers of different sequences, but selected against the same ligand, might employ similar structural and dynamical features when interacting with their ligand(s).

Aminoglycosides are broad-spectrum antibiotics that interfere with bacterial protein synthesis by binding to the A-site and distorting helix-44 of 16S ribosomal RNA (rRNA). Aminoglycoside binding causes misreads of the genetic code, premature translational stops, and bacterial death [5,6]. They also interact with other natural RNA targets including regulatory elements like the iron response element (IRE) [7], site 1 mRNA of thymidylate synthase (TS) [8], trans-activating region (TAR) [9], and Rev-responsive element (RRE) of HIV-1 [10,11], group 1 self-splicing introns [12], tRNA^Phe^ [13] and ribozymes including the hepatitis delta virus (HDV) [14,15], hammerhead (HH) [16,17], and RNase P [18]. Besides these natural nucleic acids, RNA aptamers with high affinities have been selected that bind neomycin-B [19], tobramycin [20,21], kanamycin-A [22], and kanamycin-B [23]. More recently, synthetic riboswitches for neomycin-B were obtained in yeast [24]. Although these RNAs share very little sequence similarity, their secondary structural folds are all predicted to be stem-loops.

The Protein Data Bank (PDB) database contains the 3D structures of the tobramycin [25] and neomycin-B [26] RNA aptamers, and of the neomycin-B riboswitch [27] for which aminoglycoside interactions are driven mainly by electrostatic and specific hydrogen-bonding interactions. Although originally believed to be highly specific for the selecting ligand, later evidence demonstrated otherwise. For example, some aminoglycoside aptamers have higher affinities for aminoglycoside ligands other than those against which they were selected [22,23,28,29]. Others, such as NEO1A (neo61 in ref 19), bind a variety of aminoglycosides in a buffer-dependent manner [29].

In this study, we investigated the 25-nt neomycin-B RNA aptamer (NEO2A, neo5 in ref 19), which was selected along with NEO1A. NEO2A was reported to have a similar high specificity for neomycin-B over paromomycin as NEO1A. Despite their reported similar specificity profiles, these aptamers only share 43% sequence identity. A flexible loop was identified in NEO1A as a structural feature that enables the aptamer to adapt to many aminoglycoside ligands [29]. Similar to NEO1A, we show here that the buffer composition also influences the specificity of NEO2A for aminoglycoside, although its specificity (defined by relative affinities for the aminoglycosides) differs from that of NEO1A [29]. In this study, we used a combination of biochemical, biophysical, and computational approaches to interrogate structural features that contribute to ligand specificity and affinity. Our results show that, despite minimal overall sequence identity, the NEO2A aptamer demonstrates a similar means of ligand binding as the NEO1A aptamer, which involves a preformed pocket and an adaptable pentaloop. The pentaloops are the most similar sequence spans shared by the two aptamers. As for NEO1A, we find that the NEO2A pentaloop structure varies with ligand, thereby endowing the aptamer with the ability to bind a range of ligands with high affinity. Thus, although the structure of NEO2A has not been solved at the atomic level, the structural and dynamical features by which this aptamer interacts with its ligand are the same as for NEO1A for which the structure has been determined to the atomic level. If successful with other groups of co-selected aptamers, the concept of migrating structural and dynamical features between aptamers within a group of varying primary sequence will greatly expand our ability to engineer a broad range of aptamers for improving their functionalities as sensors.

## 2. Results

### 2.1. NEO2A Binds Several Aminoglycosides with a Similar Specificity to NEO1A

Although NEO2A shares only 43% sequence identity with NEO1A (Figure 1A), both aptamers are predicted by RNAstructure software [30] to form a stem (Figure 1B). Both aptamers bound aminoglycosides (Appendix A) consisting of three and four rings (Figure 1C) with higher affinities for the four-ring neomycin-B than for paromomycin. However, the aptamers differ in their specificities for the three-ring aminoglycosides with NEO1A binding tobramycin with lower affinity compared with kanamycin-B, whereas NEO2A bound the tobramycin and kanamycin-B with similar affinities (Appendix A). These results showing similar, although not identical, affinities and specificities of NEO1A and NEO2A, suggested that these aptamers might interact similarly with aminoglycosides.

### 2.2. Molecular Dynamics (MD) Predicts NEO1A Binding to Aminoglycosides Involves Large Loop Movements

The NEO1A loop is an important feature of the interaction of NEO1A with aminoglycosides. The solved structure of NEO1A [26] shows A16 contacting neomycin-B in the binding pocket. By NMR and with 2AP-substituted NEO1A variants, we found a large impact of A16 on ligand affinity and specificity [29]. Biochemical results are consistent with the structural data that shows A16 as a “flipped-in” conformation and located over the binding pocket where it interacts with ligand [26,29]. This interaction is expected to increase ligand affinity by decreasing its k_off_. In apo-NEO1A A16 does not appear to be in this same flipped conformation [29]. This raises the possibility that the loop, and particularly A16, samples a range of conformations in the absence of ligand and clamps over the binding pocket only in the presence of ligand.

To examine the mobility of A16 in the apo-aptamer and the binary complex, we used GROMACS [31] to perform MD simulations in the presence and absence of neomycin-B. As predicted from previous studies with 2AP-labeled NEO1A, the mobility of A16 was limited by interaction with neomycin-B whereas, in the absence of ligand, A16 was free to move (Figure 2). A14, on the other hand, displayed limited motion and moved further into a stacked position in the presence of ligand.

### 2.3. A global Change in NEO2A with Ligand Binding

NEO1A and NEO2A are predicted to have pentaloop structures with almost identical base sequences (Figure 1A), which suggests that the NEO2A pentaloop may behave similarly to that in NEO1A, being mobile in the absence of ligand and pinned down in the ligand-bound complex. We tested this question using NEO2A variants modified with 2AP substitutions at each of six positions (C6, A7, A13, A14, A15, and A16) (Appendix A). These 2AP-NEO2A variants were measured for their steady-state fluorescence levels in the presence and absence of ligands (Figure 3A). Whereas the loop residue A13 moved out of a stack (increased fluorescence) upon ligand binding, A14 and A16 moved into a stack (fluorescence decrease) with ligand. There was no change in the fluorescence of residue A15. Of the 2AP variants in the bulge at positions 6 and 7, residue A7 moved out of a stack, whereas C6 was not affected by ligand binding (Figure 3A). Here, it should be noted that the replacement of 2AP for C6 is a larger structural change than 2AP for A in other positions and could involve more structural perturbation. However, these results, showing that changes in bases in the internal loop and pentaloop are affected by ligand, suggest a global structural change in NEO2A upon binding ligand.

Because the interpretation of these and other results in this study relies on the assumption that substitution by 2AP does not alter the affinity of the aptamer for the ligand, we determined (by ITC) the affinities in Buffer A for neomycin-B of a series of 2AP substituted NEO2A aptamers (Figure 3B). The substitution of A16 with 2AP, for which the affinity was 40% of the nonsubstituted aptamer, was the only substitution that resulted in a change in affinity by more than 2-fold.

To independently assess the possibility of global changes in NEO2A structure upon ligand binding, we examined 1D^1^H spectra of apo-NEO2A and NEO2A in complex with neomycin-B (Figure 3C). Although spectral assignments cannot be made without complete structural analysis, the imino hydrogen region of the 1D^1^H spectrum for the NEO2A-neomycin-B complex has the expected nine imino resonances, consistent with five Watson–Crick base pairs in the lower stem region (C1-G25, A2-U24, C3-G23, U4-A22, G5-C21) and three Watson–Crick pairs (G8-C20, C10-G18, C11-G17) and a wobble pair U9-G19. The results showing defined 1D^1^H NMR peaks provide clear evidence of a predominant structure in the apo-NEO2A and also demonstrate that the local environment of the adenosines in the loop and bulge changes when NEO2A forms a binary complex with neomycin-B.

### 2.4. Variation of Aptamer–Ligand Interactions with Ionic Environment Suggests Similar Ligand Disposition in NEO2A as in NEO1A

Despite evidence of a predominant structure for apo-NEO2A, interaction of the aptamer with ligand is expected to alter NEO2A structure as demonstrated for NEO1A (Figure 2). As well, the ionic environment profoundly influences the specificity of NEO1A for aminoglycosides [29,32]. NEO1A cradles its ligands in a structurally stable binding pocket with the loop bases responsive to buffer constituents [29]. For comparison, we tested the loop dynamics of NEO2A in Buffer F (in which these aptamers were selected), which is high ionic strength (I = 0.375 M), and Buffer A (formulated to resemble the salt concentrations in the mammalian cytosol), which is lower ionic strength (I = 0.185). As for NEO1A, the disposition of the mobile bases, established by changes in 2AP fluorescence, was influenced by ligand binding and buffer (Figure 4B). Thus, NEO2A and NEO1A both change in structure with ligand binding when monitored by 2AP reporters located in each of two positions (Figure 4B) and these structural changes are influenced by the ionic environment.

To further understand how NEO2A might interact with ligand compared with NEO1A, we examined the effects of buffer composition on ligand affinities and found that both aptamers responded with similar changes in ligand affinity in the equivalent buffer conditions (Figure 4C). As the major effect of the ionic environment will be on the portion of the ligand that is exposed to the environment, these results suggest that the orientation of ligand relative to the aptamer is similar in NEO1A and NEO2A.

### 2.5. Ligand-Specific Interactions with the Loop Bases of the NEO2A

The loop bases have emerged as being critical for ligand interaction in NEO2A as was previously found for NEO1A. If these aptamers adapt to the ligand structure in forming the binary complex rather than being driven by a more rigid apo-aptamer structure, each ligand may interact differently with the aptamer, particularly with the mobile bases such as identified in this study. Consistent with this hypothesis, the relative fluorescence changes in aptamers modified with 2AP at each of 4 positions were different for each ligand (Figure 5A).

To test the requirement for adenines in the loop for ligand binding, we determined ligand affinity for an aptamer variant lacking one adenine in the loop and, therefore, possessing a tetraloop with a string of 3 adenines compared with the 4 adenines in the NEO2A pentaloop (Figure 5B). Contrary to the expectation that elimination of adenine from the pentaloop might have the largest effect on the affinities for the ligands that caused the largest change in 2AP16 fluorescence on binding, there was no correlation between these parameters. Thus, it appears that each ligand might interact with the aptamer in a unique way, with some requiring the presence of four adenines and others not and some causing a larger change in structure in the vicinity of the pentaloop and others not. These data also support the concept of an adaptive interaction of aptamer with ligand to form the binary complex.

## 3. Discussion

With the rapid pace of aptamer discovery and their increasingly frequent applications as sensors and riboswitches and in view of the limited number of available solved aptamer structures, it seems highly unlikely that the structures of most aptamers will ever be solved to the atomic level by current biophysical approaches. Yet, understanding how nucleic acid aptamers interact with their ligands and other molecules in their environment is essential for developing effective sensors, targeting molecules (such as for therapeutics) and riboswitches. Here, we have explored the hypothesis that target ligand drives both aptamer structure and the dynamical characteristics necessary for high-affinity binding. We propose that these structural and dynamical features are similar for many aptamers that have been selected against the same ligand, regardless of the similarities of their primary structures. To test this hypothesis, we have investigated aptamers that bind aminoglycosides.

In our previous studies of NEO1A [neo61 in Reference 19], we demonstrated that its ligand specificity is a function of its ionic environment [29]. We also demonstrated that NEO1A has a structurally flexible loop and a well-defined ligand pocket. The SELEX experiment by which NEO1A was selected also yielded many other aptamers for aminoglycosides that differed from NEO1A by more than 50% in primary sequence. This raised the question of whether NEO1A is an unusual aptamer with unique properties unlikely to be found in other aptamers or if the dynamics of RNA structure observed with NEO1A can be expected in other stem-loop structures that were selected to bind the same target ligands. The underlying hypothesis to be tested was that ligand drives the selection of structural and dynamical features rather than of primary oligonucleotide sequence. To explore this question, we chose to compare NEO1A with NEO2A, a second aptamer from the same SELEX experiment but less than 50% identical to NEO1A. One reason for choosing NEO2A, of the many aptamer sequences reported that bind neomycin-B, was that NEO2A also has riboswitch activity in vivo [24]. Therefore, a better understanding of the structural basis of its interaction with ligand might have practical applications in the development of riboswitches.

High-affinity binding (Kd in the nM range) requires aptamer recognition of multiple structural features on the aminoglycoside. Here, we show that NEO1A and NEO2A bind the same subgroup of aminoglycosides but their specificities for aminoglycosides are different. The largest difference in affinities is for kanamycin B, which is low for NEO1A and high for NEO2A. For both aptamers, the fourth ring seems unimportant for aptamer–aminoglycoside interaction. For example, ribostamycin, which binds both aptamers with high affinity, similar to neomycin-B, lacks the fourth ring. This ring plays a minor role in the interaction of neomycin-B with NEO1A in its NMR structure [26]. Moreover, the NEO2A-derived synthetic riboswitches selected for neomycin-B are regulated by ribostamycin in yeast [24]. For NEO1A and NEO2A, the affinity for neomycin B is about 5 times higher than for paromomycin. As these aminoglycosides differ from each other at only one position, the amine on the 1^st^ ring was identified as crucial for aminoglycoside–aptamer specificity. With the exception of the interaction with geneticin, the binding specificity is correlated with this amino group when three-ring aminoglycosides are compared. Another chemical group with an important influence on specificity is R2 (Appendix A). Both aptamers have lower affinities for kanamycin A than B, which differ only in R2, with an amino (kanamycin B) or hydroxyl (kanamycin A) group. However, a quick scan down the list of aminoglycoside ligands shows that an amino group in the R2 position is correlated with the specificity of both aptamers except for amikacin and kanamycin-A. Therefore, specificity seems to be due to recognition by the aptamer of a pattern of chemical substituents on the aminoglycoside framework. The differences in aminoglycoside specificity between NEO1A and NEO2A show that they place emphasis on different substituent components of this overall ligand pattern.

Despite being only 43% identical in sequence, NEO1A and NEO2A share a CG clamp that defines the end of the hairpin loop and an identical pentaloop sequence with the exception of the middle base, which is a purine in both instances. Our previous NMR analysis of NEO1A and its ligand interactions along with fluorescence studies of 2AP-modified versions of NEO1A identified A14 and A16 in the loop as the most flexible structural elements of the aptamer that rearranged upon ligand binding [29]. Here, we used short (20 ns) MD simulations to analyze the NEO1A interaction with neomycin-B. Although simulations of this length are insufficient to observe the complete sampling of the apo state, short simulations of the apo aptamer do provide insights about the interactions that are stabilized in the presence of the ligand [33,34]. In this study, conformational changes in the loop and binding pocket regions were observed, it was found that A14 and A16 had mobilities that were consistent with the previous NMR and 2AP results [29].

The concordance of MD simulations, NMR and 2AP fluorescence for NEO1A for which a complete biophysical structure is available suggested a means of assessing structural models of aptamers for which the structures have not been solved by biophysical methods. An alternate approach to NMR and X-ray crystallography to reliable structural delineation could provide a path to better understanding the ligand interactions of many aptamers.

Based on their sequence identities in the region of the predicted loop we hypothesized that, as for NEO1A, loop residues in NEO2A play a critical role in aminoglycoside binding. Buffer composition, especially ionic strength, can affect aptamer affinity and specificity [29,32,35]. Consequently, we tested 2AP variants of NEO2A in buffer A (emulates eukaryotic cytoplasm) and F (high ionic strength and the conditions under which these aptamers were selected). The higher fluorescence of A13 in the binary complex compared with apo-NEO2A, which is evidence for its shifting from a stacked position to a more unstacked (hydrophilic) environment, paralleled that of A16 in NEO1A. The fluorescence changes in A14 and A16 fluorescence, which showed that they moved more into a stacked conformation, also paralleled the changes in A14 in NEO1A. In the NMR-solved NEO1A structure, A16 lies on top of neomycin-B [26]. A similar shift in A13 when NEO2A binds ligand suggests that A13 in NEO2A caps the aminoglycoside ligand in the binding pocket in a similar manner to A16 in NEO1A.

Unlike NEO1A, the secondary structure of NEO2A contains a 2-base internal bulge including the C6 and A7 residues. An increase in 2AP fluorescence was detected for the A7 position similar to A13, and no change in fluorescence was obtained for the 6^th^ residue. Therefore, we think the interaction of aminoglycosides with NEO2A is an “induced fit” mechanism involving the pentaloop and the internal bulge and a driving force from ligand binding.

Exchanging the pentaloop in NEO2A with a tetraloop by removing one of the adenines altered the affinity for some and not for other ligands. The affinities for neomycin-B and sisomicin did not change appreciably, whereas affinities for ribostamycin, paromomycin, and tobramycin were decreased. This suggests that the combination of loop and binding pocket helps to accommodate each ligand differently to form the binary complex with NEO2A.

Overall, our results show that, despite the large difference in primary sequence, the structural and dynamical features of NEO1A and NEO2A are similar, including their responsiveness to buffer components. If applicable to other groups of aptamers that have been selected against the same target, it may be possible to apply a detailed analysis of one aptamer to better understand the structural features of many others in a group of co-selected aptamers.

## 4. Materials and Methods

### 4.1. Materials and Chemicals, Buffers, and RNAs

All RNA oligonucleotides used in this study were purchased from Integrated DNA Technologies Inc. (IDT, Coralville, IA, USA) and stored at −20 °C in ddH_2_O (deionized-distilled water) until tested in the ITC or in 2AP fluorescence experiments. Sequences of the oligonucleotides used in this study are shown in Appendix A. Neomycin-B, ribostamycin, paromomycin, tobramycin, kanamycin-A, kanamycin-B, sisomicin, geneticin, netilmicin, and amikacin were obtained as their sulfate salts from Sigma-Aldrich (St. Louis, MO, USA). Cacodylic acid was purchased from Amresco (Solon, OH) and sodium cacodylate was from Sigma-Aldrich (St. Louis, MO, USA). All other chemicals were reagent grade and obtained from Fisher Scientific (Pittsburgh, PA, USA). The constituents of the low ionic strength buffer (A) and high ionic strength buffer (F) are listed in Appendix A.

### 4.2. Isothermal Titration Calorimetry (ITC)

A calibrated Microcal VP-ITC microcalorimeter (Northampton, MA, USA) was used to determine the thermodynamic parameters for aptamer aminoglycoside interactions. For each experiment, the RNA in the reaction cell and the aminoglycoside in the syringe were prepared in the same buffer and all solutions were degassed at room temperature immediately prior to use. Following thermal equilibrium at 25 °C, an initial 60-second delay and a single 1 µL titrant injection, 10 µL was injected each 150 sec for 30 serial injections into the sample cell (1.4 mL) while stirring at 310 rpm. Control experiments were performed by making identical injections of the titrant solution into buffer lacking RNA and these values were subtracted from the titration of aminoglycoside into the aptamer-containing sample cell. Data were analyzed using a nonlinear least-squares curve fitting in Origin7.0 (OriginLab Corp., Northampton, MA, USA) using the standard one-site binding model supplied with Origin7.0 (Appendix A).

### 4.3. Measurement of 2-Aminopurine Fluorescence Intensity

Variant NEO2A aptamers with 2AP replacing adenosines at positions 7, 13, 14, 15, or 16 or cytosine at position 6 were used for ligand binding assays. The binding of 10 μM aminoglycosides and 1 μM NEO2A variant in buffer A or F was monitored by 2AP fluorescence using a Cary Eclipse spectrofluorometer (Varian, Palo Alto, CA, USA). Variant aptamer alone in buffer was used to obtain a background value, which was subtracted from the raw data. For excitation and emission spectra, a quartz cell with 1 cm path length with an emission slit width of 5 nm was used with maximum excitation and emission wavelengths being 307 and 370 nm, respectively (Appendix A). The effects of aminoglycosides binding to the 2AP NEO2A variants were reported as the fold change in fluorescence at 370 nm (F370) relative to the apo-aptamer in the same buffer.

### 4.4. Molecular Dynamics Simulations

Molecular dynamics (MD) simulations were performed on the aptamer in the apo state and the aptamer ligand complex of NEO1A using GROMACS [31] with the Amber99sb all-atom force field [36]. The coordinates of NEO1A were taken from model 5 of the experimental NMR structure (Protein Data Bank (PDB) code: 1NEM.pdb) [26]. The unoccupied (apo) aptamer structure was constructed from the NMR structure by deleting the ligand, an approach used in previous studies of RNA riboswitches [33,34,37,38,39]. Force field parameters for neomycin-B were obtained from the AnteChamber Python Parser interface (ACPYPE) [40] tool, which called Antechamber [41] to parameterize the ligand and generate a topology. Initialization parameters are in Appendix A. Each aptamer was solvated in a cubic box of TIP3P water, neutralized with Na^+^ ions and energy-minimized. The minimized structures were equilibrated with 100 ps the canonical (NVT) and isothermal-isobaric (NPT) ensembles at 298 K. Non-hydrogen atoms of the aptamer were constrained during equilibration. MD production runs were carried out for 20 ns at a constant temperature of 298 K and pressure of 1 bar with a time step of 2 fs. The LINCS algorithm [42] was used to constrain bonds. All simulations were conducted using periodic boundary conditions and the particle mesh Ewald method [43] was used for long-range electrostatic interactions.

### 4.5. Statistics

Averages of two or three multiples are shown with the standard deviations shown as error bars in the graphs. The errors were calculated to consider the errors in both components when the background is subtracted or a ratio of two numbers created. Thus, the errors were calculated as the square root of the sum of the squares of the two relevant standard deviations.

## Figures and Tables

**Figure 1 molecules-24-04535-f001:**
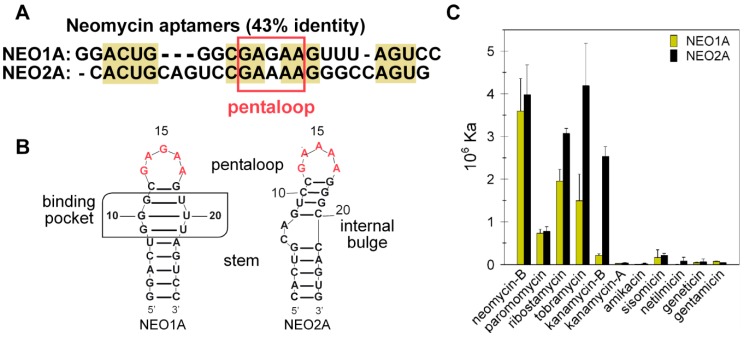
**Ligand specificity profiles for NEO1A and NEO2A.** (**A**) The aligned sequences of NEO1A and NEO2A. (**B**) 2D structural predictions for NEO1A and NEO2A. (**C**) Affinities of NEO1A and NEO2A for a series of aminoglycosides determined by isothermal titration calorimetry. The values for affinities for NEO1A were reported in Reference [29] as part of the Appendix A.

**Figure 2 molecules-24-04535-f002:**
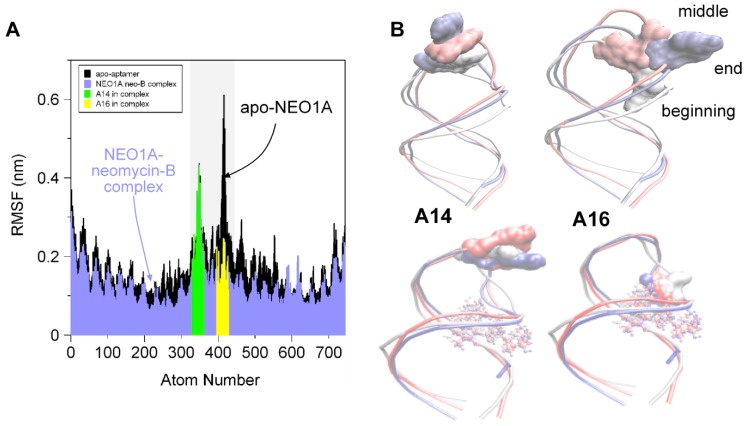
**Comparison of root mean square fluctuations (RMSFs) from molecular dynamics simulations of apo-NEO1A and NEO1A with bound ligand**. (**A**) Comparison between the RMSFs of the apo-aptamer (black) and aptamer–ligand complex (blue) simulations showed the pentaloop (light-gray rectangle) as the most flexible segment of NEO1A with the largest RMSF differences in A16 (yellow) compared with less difference in mobility of A14 (green). (**B**) Representative frames from the MD simulations for beginning (white), middle (red), and end (blue) are overlaid and are shown both for A16 and A14 movements.

**Figure 3 molecules-24-04535-f003:**
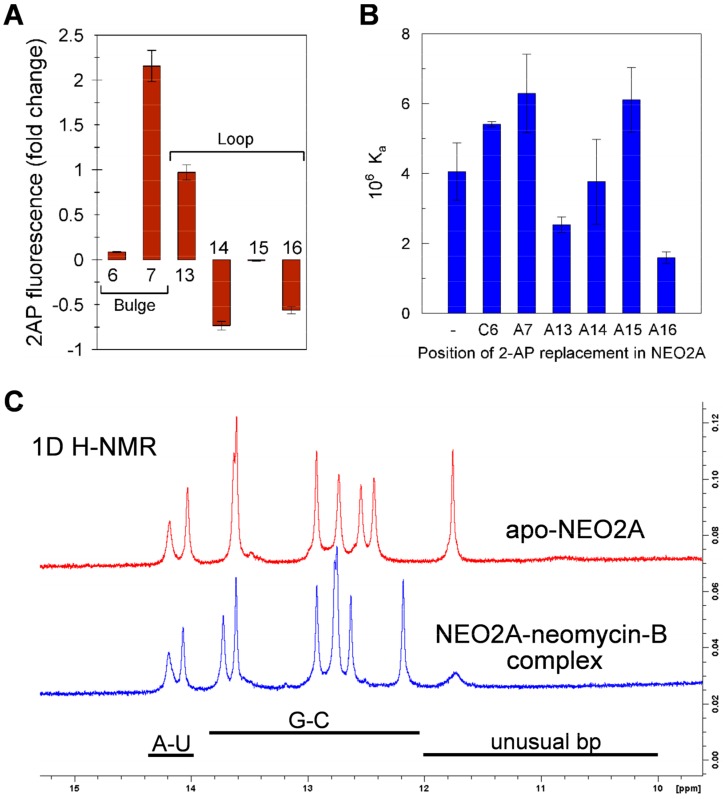
**Independent motions of the loop bases in NEO2A.** (**A**) The difference between the apo-aptamer and the neomycin–aptamer binary complex in fluorescence of 2AP located in the identified positions in NEO2A. Each test involved an aptamer with a single 2AP replacement for the base in the identified position. The fold change in 2AP fluorescence due to neomycin-B binding was calculated as [Fl(aptamer–ligand complex)/Fl(apo-aptamer)]-1. (**B**) The effect of base substitution on the affinity (K_a_) of NEO2A for neomycin-B determined by isothermal titration calorimetry (ITC). The ratio of the K_a_ determined for the 2AP substituted aptamers over K_a_ for the aptamer containing the natural base in the same position. (**C**) 1D H-NMR of 450 μM NEO2A in the absence (upper red trace) or 414 μM NEO2A in the presence (lower blue trace) of 500 μM neomycin-B.

**Figure 4 molecules-24-04535-f004:**
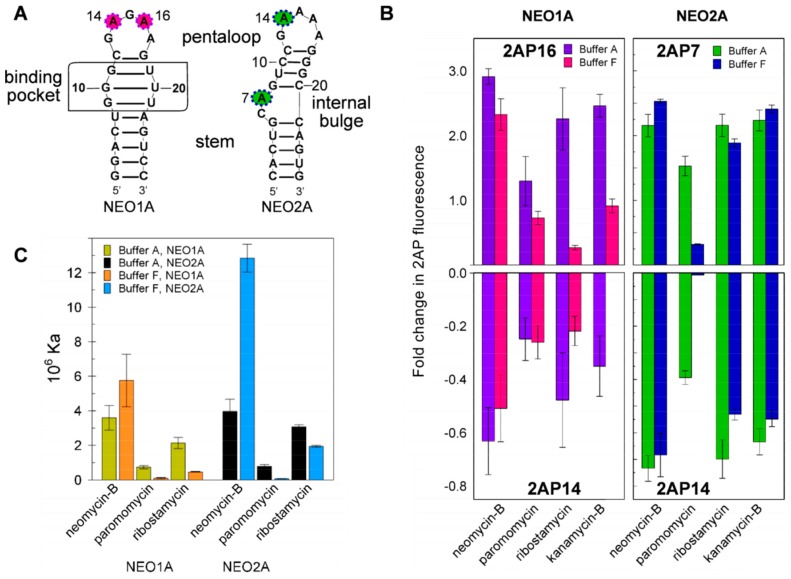
**Changes in 2AP fluorescence due to ligand binding are sensitive to the environmental salts and do not correlate with aptamer ligand affinities**. (**A**) Proposed 2D structures of NEO1A and NEO2A established by M-Fold with the colored circles identifying the two loop bases in NEO1A (A14 and A16) and one loop base (A14) and a bulge base (A7) in NEO2A that were replaced by 2AP for producing the results in B and C. (**B**) The fold change in fluorescence due to ligand binding was calculated as in Figure 3 for NEO1A and NEO2A variants containing 2AP in positions 14 and 16 or 7 and 14, respectively. (**C**) Affinities (K_a_) for a series of aminoglycoside ligands determined for NEO1A and NEO2A determined by ITC in two buffers A and F.

**Figure 5 molecules-24-04535-f005:**
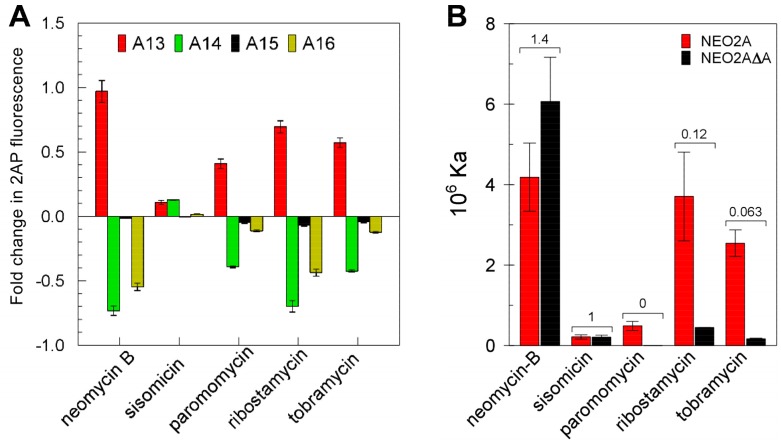
**Impact on ligand binding of the loop bases in NEO2A.** (**A**) The differences in fluorescence of 2AP between apo-aptamer and binary complexes, when 2AP was substituted separately for each A in the loop of NEO2A. (**B**) The affinities (Ka) for various aminoglycosides of NEO2A compared with NEO2AΔA (missing one A in the loop). The fold change in fluorescence due to ligand binding was calculated as in Figure 3.

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
