# Peer review of "Common Secondary and Tertiary Structural Features of Aptamer–Ligand Interaction Shared by RNA Aptamers with Different Primary Sequences"

_molecules, 2019, doi:10.3390/molecules24244535_

Round 1

Reviewer 1 Report

The manuscript „Common secondary and tertiary structural features of aptamer-ligand interaction shared by RNA aptamers with different primary sequences“ by Ilgu et al. describes a comprehensive biochemical and biophysical comparison of two neomycin binding aptamers. Such analyses are crucial for understanding RNA-small molecule interactions and thus interesting to a broad readership. I would therefore recommend publication in Molecules after small modifications:

Major comment:

Page 3, line 136: Should the NMR spectrum of the NEO2A-neomycin-B complex not contain ten imino resonances? Five for the lower stem and five for the upper stem (three for the three WC pairs and two for the GU pair)? The label in Figure 3B also indicates that four A-U base pairs are detected. However, only two should be formed. Please comment on these points.

Minor comments:

Fig S1: Gentamicin is missing. Figs S2 and S3 are not referred to in the main text. Supplementary: Please show representative ITC data for all aminoglycosides and ITC dat and fluorescence scans for all 2-AP variants The affinities of the 2-AP variants should be shifted from Figure 5 to Figure 3. These are very important controls and should be shown at the first introduction of the 2-AP variants. While reading the manuscript, I kept wondering if this control had been done. Please indicate the number of measured replicates directly in the figure legends. Page 8, line 254: The neomycin riboswitch was not “derived” from NEO2A, but from another aptamer stemming from the same in vitro selection, which a similar organization of the bulge region to NEO2A, but a different terminal loop. Please rephrase. Page 8, line 258: I do not understand the comment “…specificity is not strictly correlated with this amino group in three-ring aminoglycosides…”. With the exception of geneticin, which does not bind to the aptamers, all three-ring aminoglycosides tested contain this amino group. Thus, no clear statement can be made. Page 8, line 262: “…amino group in the R2 position is not correlated with specificity”. Similar to above. Except for amikacin and kanamycin-A, both of which are not recognized, all other tested aminoglycosides contain this amino group. So it seems to me that this amino group is important for binding, also obviously not the only determining factor.

Author Response

Dear reviewer,

thanks for your input. Your valuable comments are highly appreciated.

Best 

Reviewer 2 Report

Ilgu et al. in this manuscript has explored an interesting concept of aptamer ligand interaction where they have shown small molecule target induced change in aptamer structure plays crucial role in binding. The authors have nicely shown that two aptamers with only 43% sequence similarity (also with a significantly different internal bulge structure) can bind to the same target. They also nicely shown small molecule induced structural change in the aptamer that compensates the structural dissimilarity of the native aptamer structure, suggesting an induced fit model.

However, the initial choice of aptamer, according to me, are not a good choice. The NEO1A aptamer is not specific to a particular aminoglycoside and binds to a broad range of aminoglycosides. Therefore, the concept of structure dependent induced fit model is questionable. The aptamers used here are promiscuous and therefore, may act as a general scaffold for aminoglycoside binding. 

To address there claim (and the hypothesis is really interesting), the authors should use aptamers that are specific to a single target and then do the experiments to prove their hypothesis.

I think the authors really need to use highly specific aptamers to support their claim. 

Author Response

(The authors gave the same response as above.)

Reviewer 3 Report

Dear editor,

I carefully read the manuscript entitled "common secondary and tertiary structural features of aptamer-ligand interaction shared by RNA aptamers with different primary sequences". This article relates the structural comparison, through experimental and theoretical investigations, of two RNA aptamers able to bind the same aminoglycoside ligand. In my opinion, this work fits the scope of "molecules" journal and provides interesting analyses on two distincts RNA sequences. However, I raised several remarks which should be answer to definitely accept this manuscript for publication in your journal.

- A conclusion is missing. This concluding part should summarize the main results and provide future expected prospects.

- How the 43% of sequence identity between RNAs have been computed?

- The authors refer to the use of RNA aptamer as biosensor. However, in this context, grafting onto a surface may triggers structural variations on the 3D structures of RNA as demonstrated by M. Ruan (J. Phys. Chem. B 2017, 121, 4071−80). In the light of this work, I think this point could be discussed by the authors.

- Even if ref19 is in open-access, I think that the molecular scheme of the ligand must be provided (with the correct chirality).

- The authors used antechamber to create the required topology and parameters for the ligand. What type of atomic charges computation was chosen?

- In my opinion, the main information provided by the authors is that ligand induces a specific structural organization of the RNA, "whatever is its sequence". To my opinion, this sounds correct to my feeling and knowledge of aptamers. However, with only the case of neomycin-B this final conclusion is still elusive. Therefore, I suggest that the authors take care to not so much extrapolate their results and their conclusions according to the fact that they studied only the case of neomycin-B.

Author Response

(The authors gave the same response as above.)

Round 2

Reviewer 2 Report

The authors are nicely addressed the concern raised and now I think the manuscript can be accepted for publication